# A Study of the Structural Organization of Water and Aqueous Solutions by Means of Optical Microscopy

**Tatiana Yakhno [1,2,]*** and **Vladimir Yakhno [1,2]**

1 Federal Research Center Institute of Applied Physics of the Russian Academy of Sciences (IAP RAS), 603950 Nizhny Novgorod, Russia; yakhno@appl.sci-nnov.ru
2 N. I. Lobachevsky State University of Nizhny Novgorod (National Research University), 603950 Nizhny Novgorod, Russia
* Correspondence: yakhta13@gmail.com; Tel.: +7-831-436-85-80

**Abstract:** The structural organization of water and aqueous solutions under an optical microscope in a layer with a thickness of 8 μm was investigated. It is shown that under room conditions water (including "ultrapure" water) and aqueous solutions are microdispersed systems. The revealed effect does not depend on the properties of the substrate (texture, hydrophilicity/hydrophobicity) and is an inherent property of the liquid. The disperse phase is based on contrasting micron-sized formations located in the center of low-contrast homogeneous spheres observed in a layer with a thickness of the order of the diameter of the observed structures. They form loose millimeter-sized associates in the liquid phase. When the water is boiled, the associates become disordered, but the dispersed phase is preserved. An increase in the ionic strength of the solution is accompanied by coacervation of the dispersed phase. When the liquid part of the water evaporates, the microdispersed phase remains on the substrate. The central particles begin to grow and take on the form of crystals. On the basis of the literature data and their own research, the authors believe that the structures are sodium chloride microcrystals surrounded by a thick layer of hydrated water. Possible ways of salt penetration into the aquatic environment are discussed.

**Keywords:** microstructure of liquids; water-salt units; salt microcrystals; hydrated shells; coacervates; self-assembly; air pollution

---

**Instead of An Epigraph**

" . . . a scientist must also be absolutely like a child. If he sees a thing, he must say that he sees it, whether it was what he thought he was going to see or not. See first, think later, then test. But always see first. Otherwise you will only see what you were expecting. Most scientists forget that. I'll show you something to demonstrate that later. . . . You can't possibly be a scientist if you mind people thinking that you're a fool."

Douglas Adams.

"So long, and thanks for all the fish" ("The Hitch Hiker's Guide to the Galaxy" #4)

## 1. Introduction

Everyone knows that water is a universal solvent. This means that every substance that has contact with water leaves traces of its presence in it. These include the elements of the material of the vessel in which water is stored, and aerosol microparticles surrounding the vessel, and traces of the chemical composition of the devices in which it was purified, and dissolved gases. Classical science considers water exclusively as a more or less loose framework of $H_2O$ molecules, whose molecular clusters are capable of rearranging in nanometer space and picosecond time [1–7]. Small-angle X-ray

scattering (SAXS) was used to demonstrate the presence of density fluctuations in ambient water on a physical length-scale of 1 nm [8]. Within the framework of the present paper, we shall not touch upon the questions of the molecular dynamics of water and the diagram of its states over a wide range of temperature and pressure changes. A large amount of information is devoted to these problems [9–14]. We intend to pay attention to the structural organization of water under ordinary (room) conditions, as can be observed under an optical microscope.

For the first time, to our knowledge, the structural heterogeneity of aqueous dispersions was revealed by Japanese researchers at the end of the last century [15]. Using a laser confocal microscope to observe the suspension of latex microparticles in water of high purity, they found free long-lived voids. The mechanism of their formation was attributed to the mutual attraction of like-charged particles through ions carrying an opposite charge [16–18]. Particles, forcibly placed in such voids, showed mobility limitations compared to free particles [19]. Over time, the voids disappeared, and a colloidal crystal was formed in the dispersion. It was shown that the distance between the particles decreased with increasing salt concentration. But, after exceeding a certain threshold of ionic strength, the colloidal crystal thawed [16]. In our earlier research, we described the existence of spherical structures in the liquid phase of coffee, their periodical occurrence, growth, destruction, and re-emergence, which agreed with fluctuations of physicochemical parameters of the system [20,21]. We assumed that these spheres are nothing but liquid crystal water which forms thick shells around hydrophilic colloidal particles. Agglomerates of such crystal water spheres look like voids inside colloidal dispersions under a confocal scanning laser microscope. Phase transitions between free and bound (liquid crystal) water in the bulk of colloidal systems are the pacemaker of the fluctuations of the system's physicochemical properties. These changes are regulated and coordinated by the value of osmotic pressure. The existence of self-oscillatory processes was confirmed by a mathematical model [21]. At that time we could not guess that the same spheres are present in water too. The later discovery of the same spheres in water prompted us to perform a detailed study of their possible origin.

The bubbston concept of the structural heterogeneity of water and aqueous solutions has been actively developed since 1992 [22–29]. Experiments on laser modulation-interference phase microscopy and on the scattering of laser radiation have revealed the presence of macroscopic particles in aqueous media cleaned of solid impurities. After degassing the solutions by passing helium through them, structural inhomogeneities were not observed [23]. The experimental data obtained for distilled water and aqueous solutions of NaCl allowed the authors to interpret these particles as clusters of air nanospheres stabilized by ions—bubbstons [29]. Based on the results of the simulation, it was assumed that the clusters have a characteristic radius of $\cong 0.5$ μm and a fractal dimension in the range of 2.5–2.8. Based on the mean scattering cross section of these clusters, their concentration in distilled water and in an aqueous 0.8 M NaCl solution was estimated as $\cong 10^3$ cm$^{-3}$ and $\cong 2 \times 10^6$ cm$^{-3}$, respectively [30]. The authors of paper [31] by the method of small scattering angles in bidistilled water found optical inhomogeneities-clusters-in the size of 1.5–6.0 μm. However, the nature of these inhomogeneities remained unknown. At the same time, the existence in the water of giant clusters with sizes from 10 to hundreds of microns was proved by means of IR spectroscopy [32], laser interferometry [27,33], small-angle light scattering [34,35], dielectrometry, and resonance method [36]. The authors of the research express various assumptions about the mechanism of formation of the revealed structures. In our opinion, in these works and our observations we are talking about the same structures visible in a light microscope. In this article, we have presented facts that testify to the validity of our reasoning. The tasks of our work included the study of the microstructure of water and aqueous solutions under an optical microscope to detail the results we obtained earlier [20,21]. We could not find such information in the available literature. The observed phenomenon is undoubtedly closely related to the physicochemical properties of water, the anomalies of which have not so far been satisfactorily explained [37,38].

## 2. Materials and Methods

The experiments were carried out under laboratory conditions at T = 22–24 °C, H = 73–75%. Distilled water was used for microscopic observations (TU 2384-009-48326337-2015, specific electrical conductivity 4.5 μS/cm, pH 7.0), tap water (specific conductivity 550 μS/cm), ultrapure water (OST 34-70- 953.2-88, specific conductivity 0.04–0.05 μS/cm, pH 5.4–7.0), mineral natural drinking water "Seraphim Dar" (mineralization 0.05–0.12 g/L). In addition, aqueous solutions of freeze-dried instant coffee "Nescafe Gold", 125 g/100 g of hot tap water after cooling to room temperature, and dried smears of this solution were examined. A dry white wine "Chardonnay Tamani" was also studied among other liquids. The studies were performed under a Levenhuk microscope with a video camera coupled to a computer using the ToupView program. In the work, the microscope slides and cover glasses of ApexLab (Russia) production were used, with dimensions (25.4 mm × 72.2 mm × 1 mm) and (24 mm × 24 mm × 0.6 mm), respectively, Petri dishes d = 35 mm (polystyrene, sterile, MiniMed, Russia). Only new glass and plastic dishes were used, without additional processing. Samples of liquids in the form of droplets in a volume of 5 μL with a Sartorius microdoser (Biohit) were applied to the substrates and covered with a coverslip. The crystals of muscovite were also used in the work. A thin crystal plate was chipped from the end and a drop of water was introduced into the formed slit capillary, spreading out in the form of a thin film. The thickness of the water film in the preparations was evaluated to be ~8 μm using geometry rules. Microscopic preparations were observed in transmitted light. NaCl of the brand "hch" ("Reaktiv", Russia) was used to prepare saline solutions. Some part of the solutions was centrifuged at 5000 rpm for 20 minutes, after which smears were prepared from the bottom of the centrifuge, air dried, and microscopically tested without the use of a cover glass. A fine graphite powder, prepared in the laboratory, was dispersed over the glass and plastic surfaces to contrast the relief. The surface of the slide was also examined with a scanning white light microscope ZYGO NewView 7000 (Objective: 50x mirau, Camera Res: 0.110 μm, Image Zoom: 2×).

## 3. Results

### 3.1. The Essence of the Phenomenon

Water and aqueous solutions at room conditions are dispersed systems in which microparticles with a hydrophilic surface have a significant hydrated shell and form a dispersed phase, and hydrophobic microimpurities combine to form fractal clusters (Figure 1).

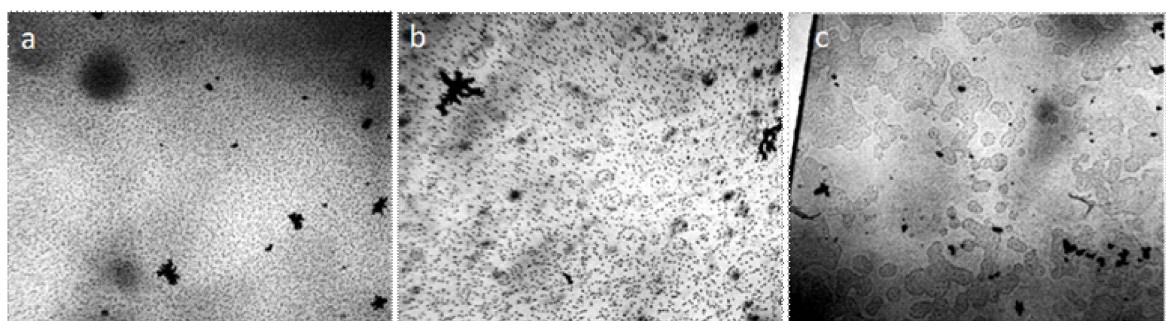

**Figure 1.** Structure of water and aqueous solutions under an optical microscope: (**a**) distilled water; (**b**) instant coffee; (**c**) white dry wine. The thickness of the liquid layer, bounded by the substrate and cover glass, is ~8 μm. The structural unit of the hydrophilic dispersed phase is visible as a regular circle with a dark particle in the center, hydrophobic—in the form of a dark cluster. The width of each frame is 3 mm.

Centrifugation of liquids (5000 rpm, 20 min) did not lead to the formation of a pronounced precipitate, but the smears, prepared on the glasses from the bottom fraction of the centrifuge, contained

agglomerates of coalesced round structures with a dark particle in the center (Figure 2). The size of the agglomerates could reach several millimeters. We found that visible circular structures of 10–15 microns in size with a dark microparticle in the center are present both in water (distilled, mineral, tap water) and in aqueous solutions. They have a greater light-scattering capacity compared to liquid water; they are quite plastic and easily stick together under deforming influences [39]. As our experiments have shown, heating the solutions to 100 °C does not lead to the dissolution of the structures. By the results of centrifugation, it can be asserted that their density slightly exceeds the density of water. Extraction of soluble organic substances from the aqueous solution of coffee by hexane and ether does not lead to the disappearance of structures, which confirms their inorganic nature [39]. Using a chromatography-mass spectrometric analysis of a sample of a dried coffee solution, it is shown that the mass of the liberated water relative to the mass of the analyzed sample is nonlinearly related to the thermal desorption temperature and is described by a third-order polynomial: $y = 8E - 0.6x^3 - 0.0043x^2 + 0.8142x - 44.191$, when $R^2 = 1$ (Figure 3). It can be seen that in the temperature range of 200–300 °C the amount of water released from the sample sharply increases, which usually occurs when the crystal hydrate complexes are destroyed.

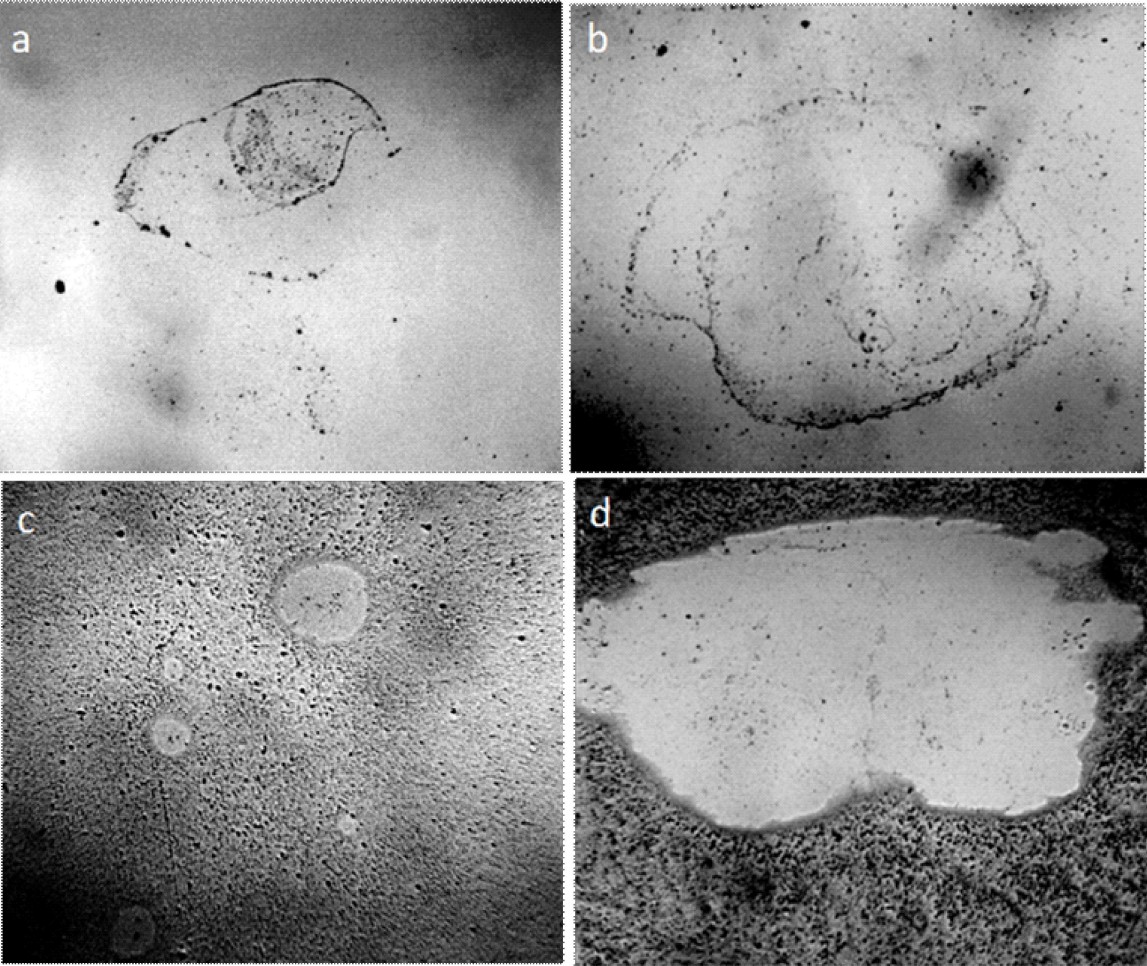

**Figure 2.** Fragments of dried out smears of the investigated liquids: (**a**) Distilled water; (**b**) white dry wine; (**c**,**d**) are instant coffee. Agglomerates from the coalesced round structures are shown in Figure 1. The width of each frame is 3 mm.

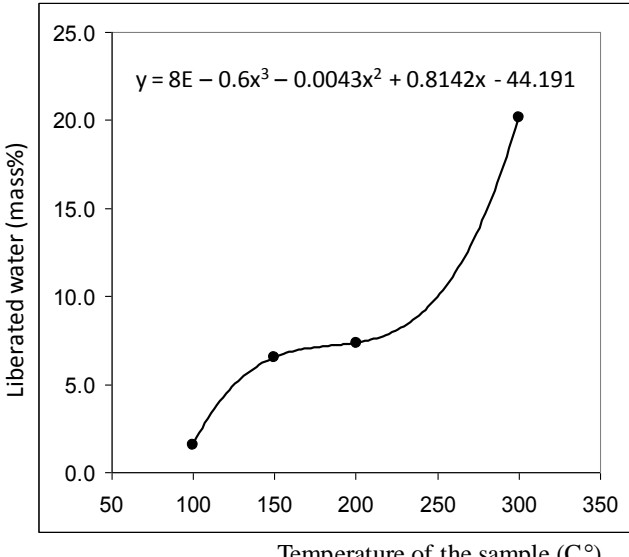

**Figure 3.** Results of chromatography-mass spectrometric analysis of a sample of a dried coffee solution for water content [39]. (Table S1 in Supplementary Materials).

In the IR spectrogram of water vapor at this temperature, in addition to the band characteristic of ordinary water, a band with two peaks, 1595 and 1400 cm$^{-1}$, was marked [39]. It is noteworthy that this particular absorption was peculiar to the legendary "polywater" (water-II) [40], which also evaporated at a temperature above 200 °C. "Polywater" was "a transparent viscous liquid with a density of 1.4, a refractive index of 1.48, nonvolatile at room temperature, with a linear expansion in the temperature range of −40 to −60 °C, which transforms into a glassy state at −40 °C due to an increase in viscosity" [41]. If the analogy between the "polywater" and the hydrate shells of microparticles observed by us is correct, then the desire to obtain "polywater" under sterile conditions was futile [42,43] and its detection in a biological fluid (sweat) is quite a natural phenomenon [44].

The results of the experiments suggest that the plastic structureless mass surrounding the central particle is bound hydrated water. What is the nature of the central microparticle that holds a significant mass of hydrated water? The observation of water drying in the open air can answer this question. Figure 4 shows the growth of crystalline structures in the drying film of tap water, the volume of which is 1 ml, (electrical conductivity is 550 mS/cm) after 5, 7, and 14 days from the beginning of evaporation in the open air. After 5 days, liquid microdroplets of water can be still observed along the edge of the dried film; microparticles holding these microdroplets can be seen within each drop of water (Figure 4a,b). After 7 days, the growth of microparticles is noted (Figure 4c), and after 14 days the crystalline nature of these particles does not cause doubt any more (Figure 4d). An x-ray analysis confirmation was obtained that these are NaCl crystals [45] (Supplementary Materials, S1).

When the distilled water drop dries on the hydrophobic polystyrene substrate (the bottom of the plastic Petri dish), the hydrophilic dispersed particles concentrate in the center of the drop (Figure 5a), and the tap water, poured into this dish in 2 mm layer, turns in 4 weeks into a structured hydrophilic dispersed phase (GDP) film coated with druses of crystals (Figure 5b). We were surprised when convinced that the structure of "ultrapure" water is no different from the structure of distilled and even mineral water (Figure 6).

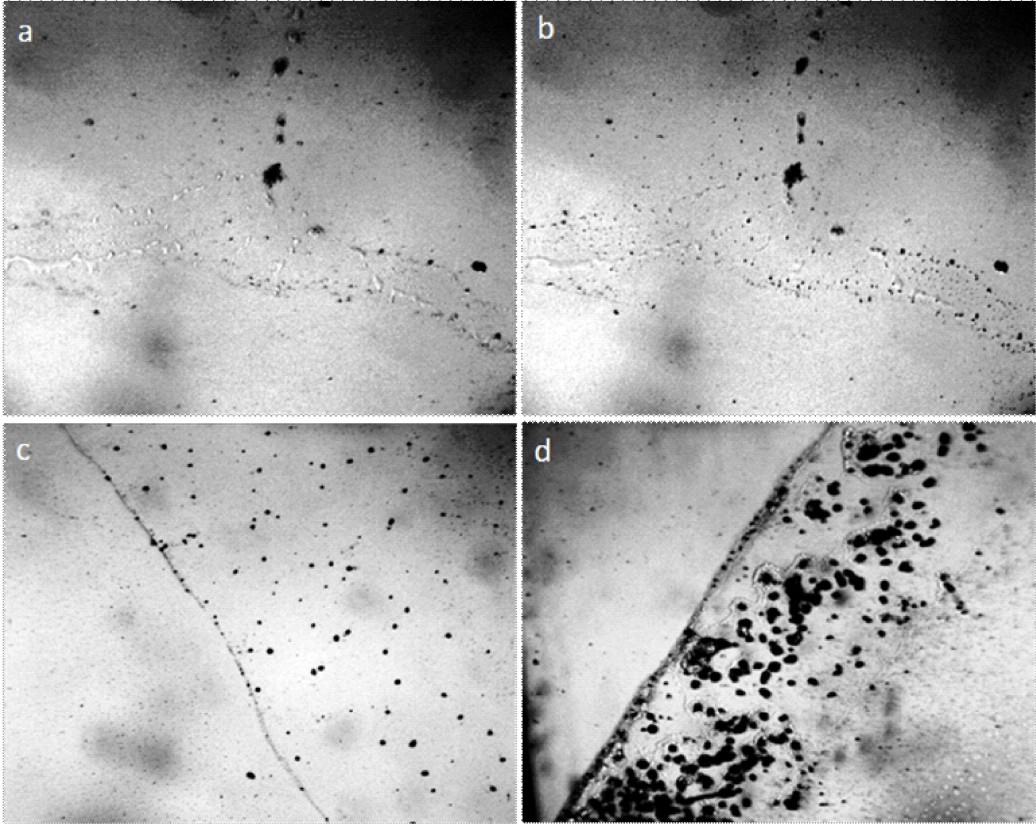

**Figure 4.** The edge of tap water film dried on the glass in the open air: (**a**) after 5 days (focus on the upper boundary of liquid water droplets; (**b**) focus on water-holding particles); (**c**) crystal size after 7 days of drying; (**d**) the size of the crystals after 14 days from the moment when the water was placed on the glass. The width of each frame is 3 mm.

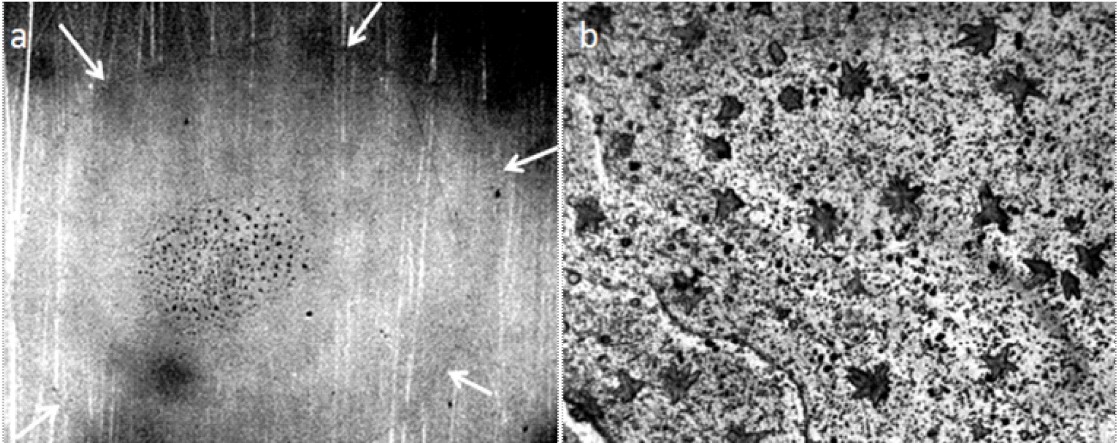

**Figure 5.** Traces of dried water on a plastic substrate in the open air: (**a**) a dried drop of distilled water (arrows indicate the outer boundaries of the drop), the center is concentrated hydrophilic dispersed phase; (**b**) a tap water film poured into a hydrophobic Petri dish with a layer of 2 mm, after 4 weeks of drying. Druzes of grown crystals are located on a GDP film. The width of each frame is 3 mm.

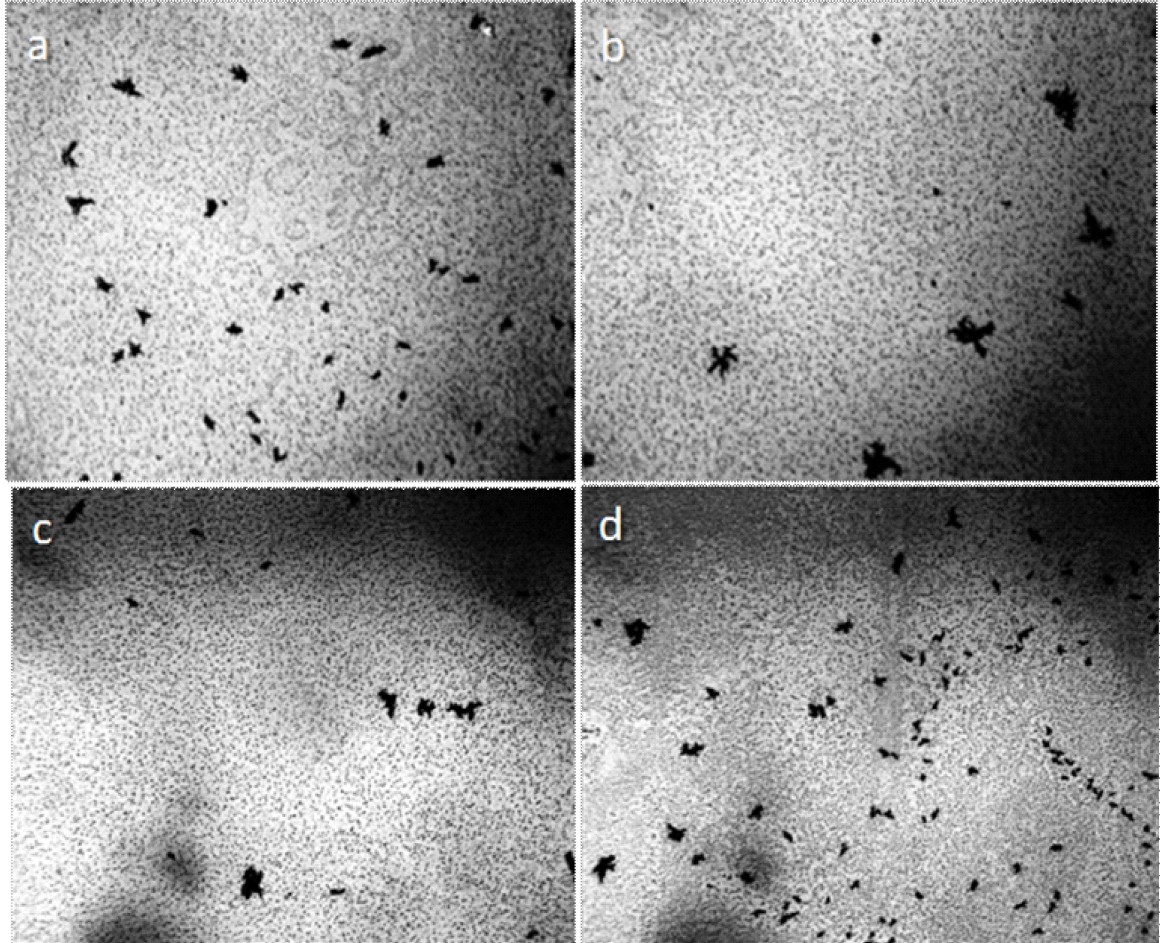

**Figure 6.** The structure of "ultrapure" water (**a**,**b**) immediately after depressurization the container essentially corresponds to the structure of distilled (**c**) and mineral (**d**) water. The width of each frame is 3 mm. The liquid layer thickness is 8 μm.

This fact suggests that the formation of microstructures in water is its inherent property.

### 3.2. Glass Surface under Optical Microscope

Considering the water on the surface of the slide, we could notice that the dry glass surface is completely covered with flat circular formations having a dark particle in the center. This pattern did not disappear when treated with water and alcohol, or after trying to wipe it with a napkin. However, this pattern was not visible on the surface of the hydrophobic plastic. Study under interference microscope showed that the "dark spots" in the center of the circles represent crystal structures with an average width of 3–5 microns and a height of 80–200 nm (Figures 7 and 8).

It is known that in real conditions the glass surface at the boundary with air is always covered by a film of water, the thickness of which depends on temperature and humidity. Removal of the film requires heating the glass above 200 °C [46].

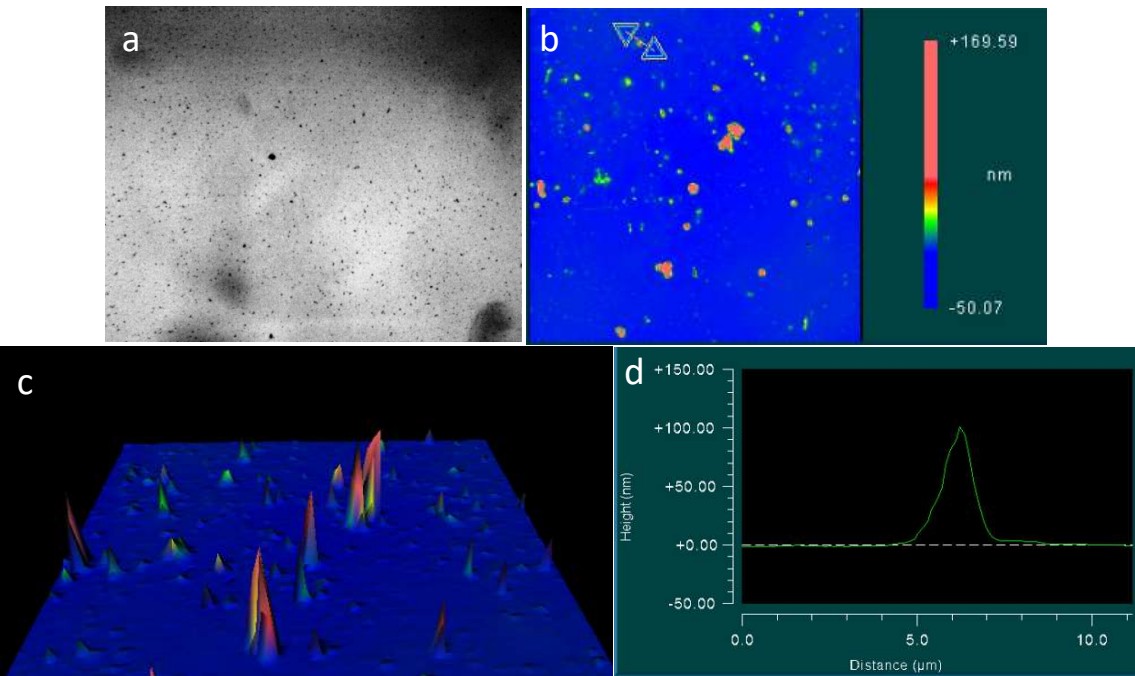

**Figure 7.** The topography of the glass surface at the glass-air boundary under the optical and interference microscopes: (**a**) optical microscopy, the width of the frame is 3 mm; (**b–d**) interference microscopy. The horizontal scale represented is in micrometers; the vertical scale is in nanometers.

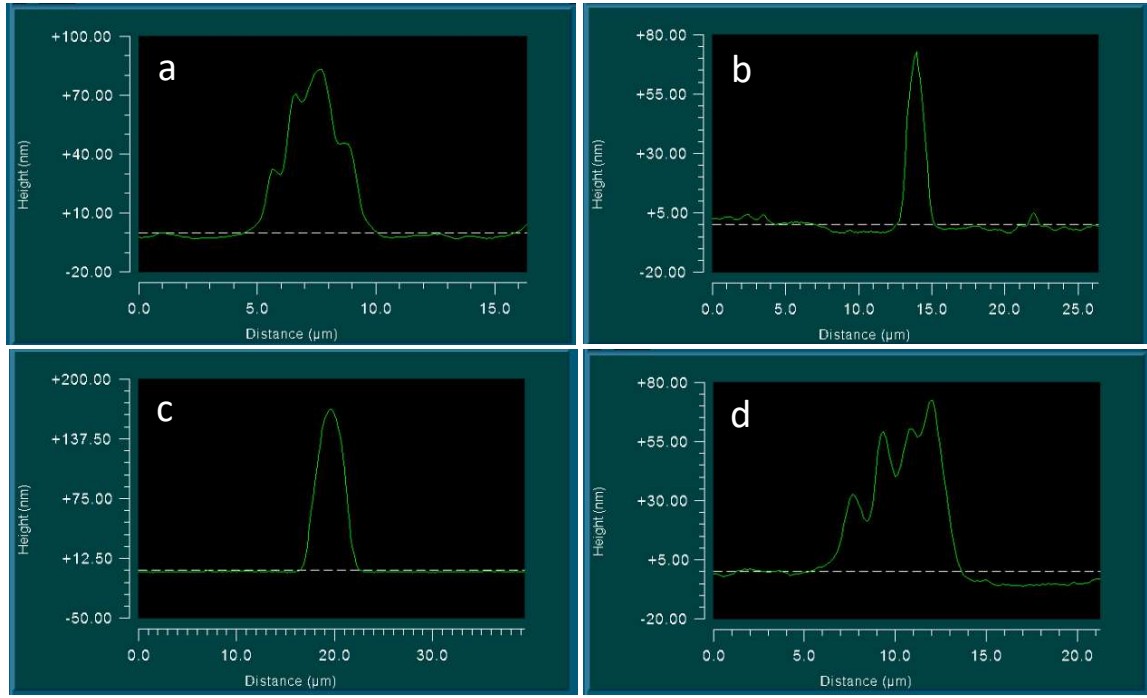

**Figure 8.** Samples of the crystal structure in the center of the circles, covering the surface of a glass slide, according to interference microscopy: (**a,b**) are types of different crystal shapes. The horizontal scale is in micrometers; the vertical scale is in nanometers.

Currently, it is possible to consider that the first layer of water molecules on the hydrophilic surface has an ice-like structure [47]. This fact is confirmed repeatedly by different research methods; the formation of an ice-like film occurs even at room temperature [47–51]. Molecules of the second and subsequent layers of the film of water are retained due to the interaction with the underlying

layers. In the review [13] the following argument was given: "The Na$^+$ ions have diameters which are comparable to that of water molecules and therefore they can substitute water molecules within the adsorbed water layer, with a minimal rearrangement of the interfacial water molecules." On the other hand, it is likely that the hydration shells of microcrystals of salts, contained in the surrounding air, have a strong affinity to the water film covering the surface of the glass. Settling on the surface, they are strongly "fused" with it through the mechanism of adhesion, thus, reducing the free energy of the macrosystem. What happens on the surface of the hydrophobic plastic? Incubation of dry glass slides and dry plastic Petri dishes in a freezer at −20 °C leads to similar results—the formation of regular icy patterns on their surfaces (Figure 9). Does this mean that the plastic surface also has similar "seed" centers of crystallization of ice from the vapor phase?

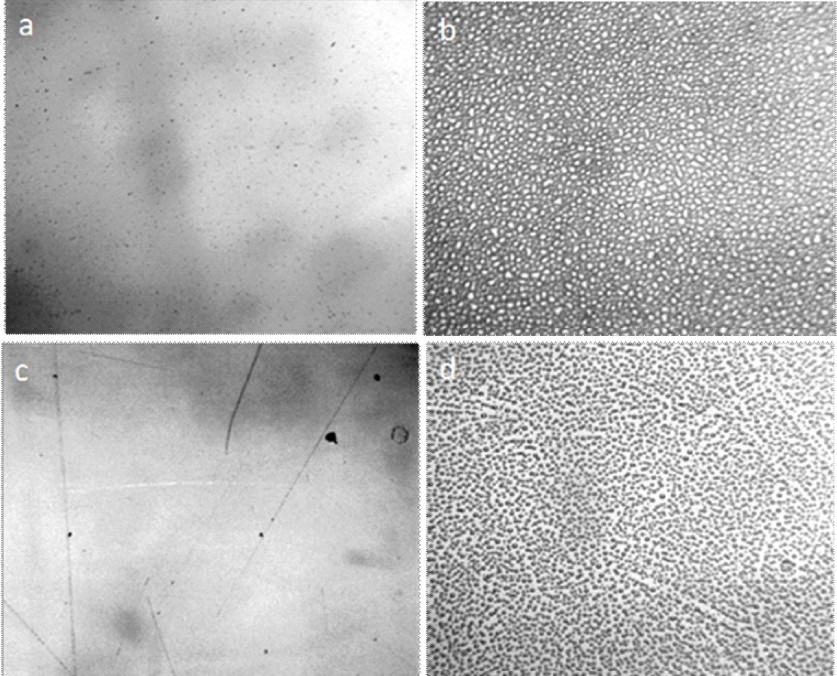

**Figure 9.** The formation of regular icy patterns on the surface of the dry glass (**a**) and dry plastic (**c**), after incubating them in the freezer at −20°C (**b** and **d**, respectively). The width of each frame is 3 mm.

To contrast the surface structure of glass and plastic we used a light airy coating of them with fine graphite powder (Figure 10).

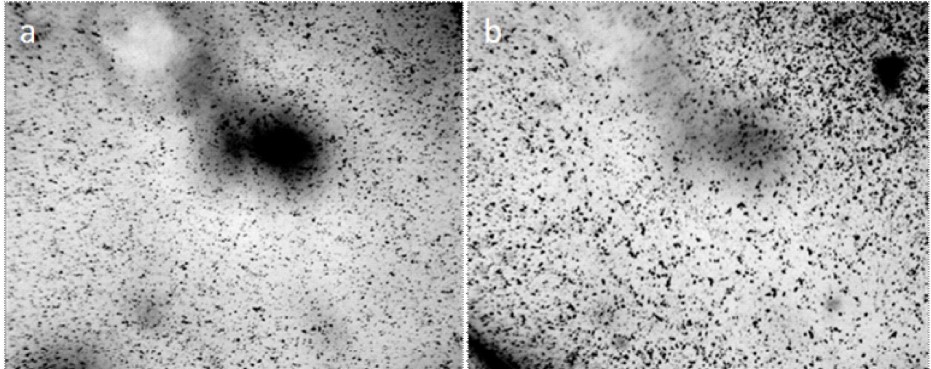

**Figure 10.** Identification of the structure of the glass (**a**) and plastic (**b**) surface via coating them with fine graphite powder. Borders of rounded structures of micron size can be seen. The width of each frame is 3 mm.

The results of the experiment showed that accumulations of hydrophilic microstructures, like aerosol contaminants, are present on the surface of hydrophobic plastic.

To exclude the influence of surface contamination on water structuring, we filled in a freshly prepared slotted capillary of the muscovite crystal (mica) with it. In the absence of surface dust contaminants, the structure of water in a thin layer was the same as when it was observed in a system of two glasses (Figure 11).

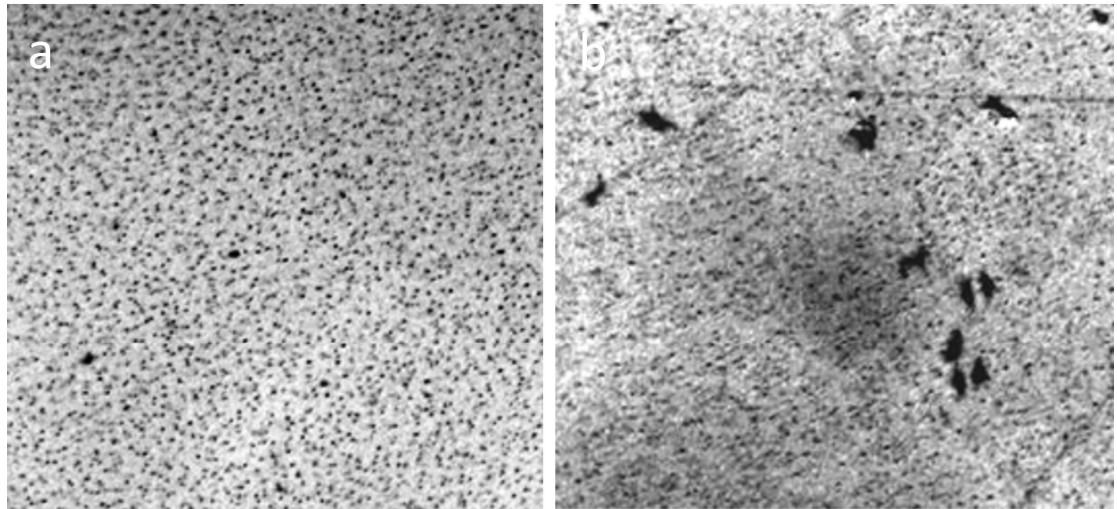

**Figure 11.** Structure of a thin layer of water (~8 μm) between the objective and coverslip (**a**) and in the slit capillary muscovite (**b**). The width of each frame is 3 mm.

In order to exclude the influence of the contacting surfaces on the water structures, we also observed a sample of water placed in a round hole (pore) in a plastic plate. Thus, the water was not restricted from above and below by any surfaces, but was located in the form of a suspended drop in the hole, being held there by capillary forces (Figure 12).

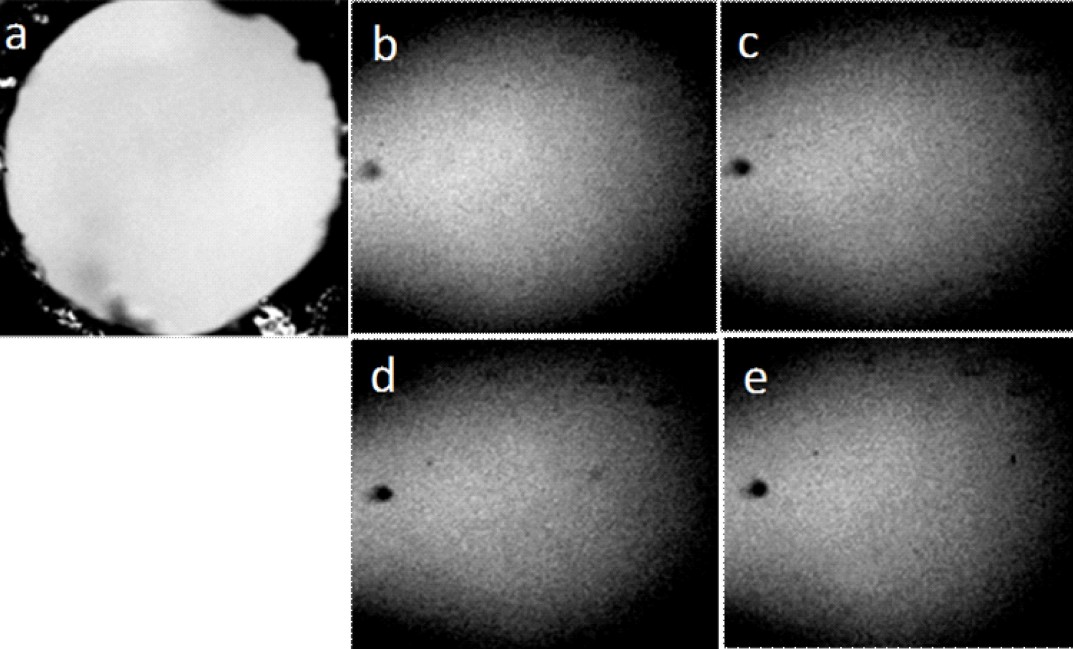

**Figure 12.** Structure of water placed in the hole in the plastic plate (pore the diameter of which is 3 mm): (**a**) pore not filled with water; (**b**–**e**) water in the pore, four consecutive frames. The width of each frame is 3 mm.

The presented material demonstrates that the structuring of water at the micro level under room conditions is not an artifact. The structural unit of the dispersed phase is a NaCl microcrystal (average diameter is 3–5 μm) surrounded by a thick layer of hydrated water (average diameter is 10–15 μm).

### 3.3. Coacervation of the Dispersed Water System. Formation of a New Phase

In water preparations, enclosed between the objective and cover glasses, it is often possible to observe the process of coacervation—combining small particles of the dispersed phase into larger structures isolated from the dispersion medium (Figure 13a,b).

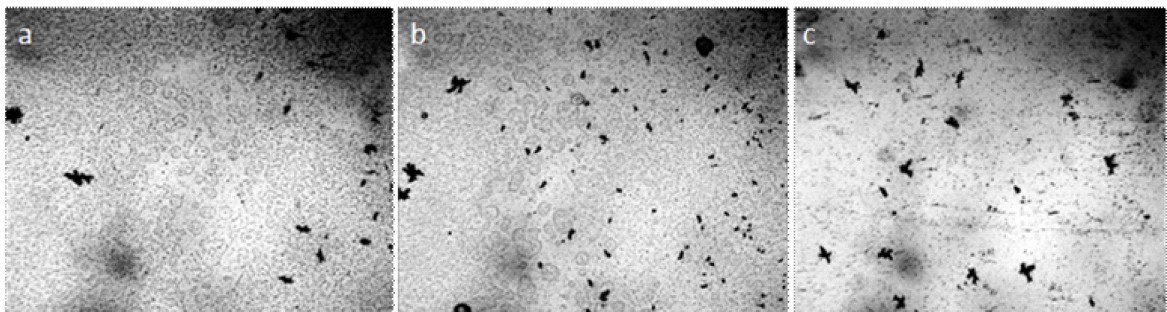

**Figure 13.** Coacervation of the dispersed phase of water in thin films (~8 μm), enclosed between the objective and coverslip: (**a**) distilled; (**b**) tap water; (**c**) is (**b**) after boiling. Optical microscopy. The width of each frame is 3 mm.

The procedure of boiling water led to disordering of its coacervate structure to a large extent (Figure 13c). The effect of cooling on the coacervate structure of the coffee solution is shown in Figure 14. Reducing the size of structures with cooling coffee to +4 °C is associated with an increase in the density of liquid water at this temperature.

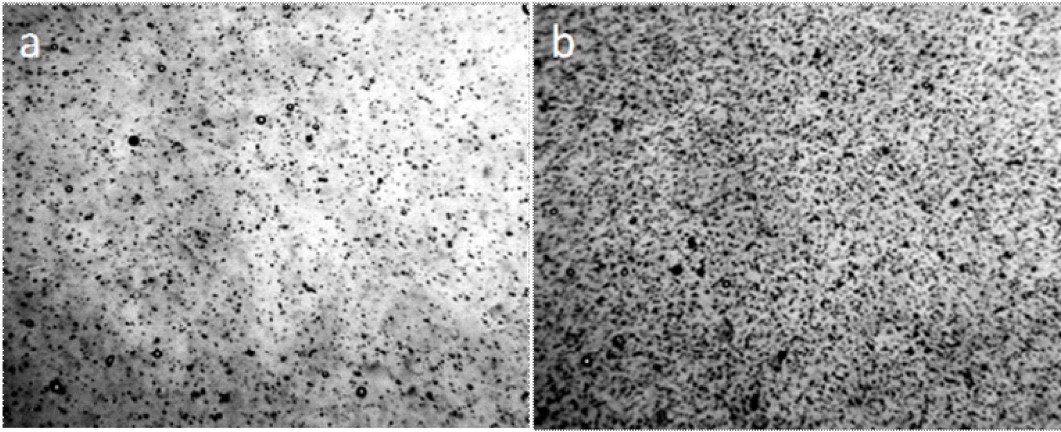

**Figure 14.** Microscopy of the open surface of an aqueous solution of coffee, poured into a Petri dish, at different temperatures: +20 °C (**a**) and +4 °C (**b**). Increase in density and decrease in the size of coacervate structures upon solution cooling. The width of each frame is 3 mm.

As is known, the factors leading to agglomeration (lowering of temperature, increasing concentration, and desolvation of particles) are, simultaneously, factors that lower the osmotic pressure and increase the viscosity of polymer solutions, and hence the factors of their latent coagulation [52]. The change in the structure of water as the ionic strength of the solution increases is shown below (Figure 15).

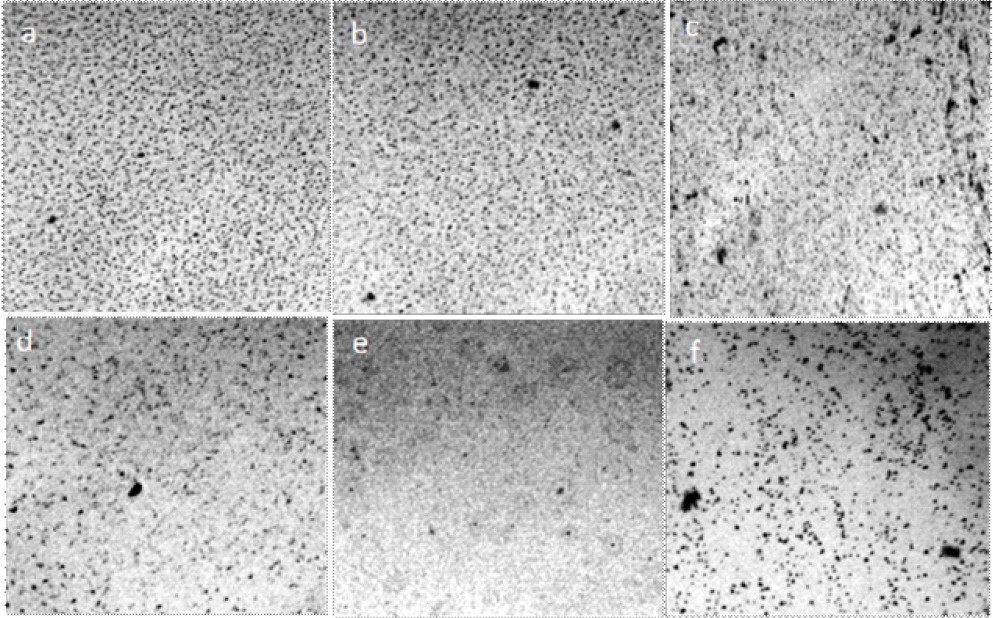

**Figure 15.** Structure of liquids in thin films (~8 μm), enclosed between the objective and coverslip: (**a**) distilled water; (**b**) 1% NaCl solution; (**c**) 2% NaCl solution; (**d**) 3% NaCl solution; (**e**) 4% NaCl solution; (**f**) is a saturated NaCl solution. Optical microscopy. The width of each frame is 1 mm. The liquid layer thickness is 8 μm.

One can observe how larger spherical associates are formed with increasing ionic strength. The structure of these associates becomes clearer after the evaporation of liquid water through the open boundaries between the cover glass and the substrate, accompanied by an increase in the ionic strength of the solution (Figure 16). The structures remaining on the slide are coacervates containing primary hydrated particles that retained their independence (Figures 17 and 18).

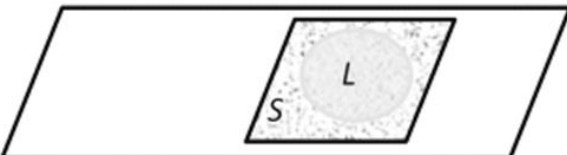

**Figure 16.** Area of microscopic observations (S) between the objective and coverslip after the partial evaporation of the liquid phase (L) through the air-bound boundaries.

In the course of this work, it was shown that the most common and most studied liquid—water—has a clear structural organization at the micro level. The information is obtained with a conventional light microscope and is available to each researcher for independent verification. We have also shown that the structurization of water is not determined by the influence of the surface properties of the substrate. According to our data, microcrystals of sodium chloride act as the organizers of the structure of water at the micro level, their source of origin requires additional study. Bunkin N.F and Bunkin F.V. [22] believe that after primary water purification (distillation, ion-exchange sorption), inorganic ions remain mainly in water; after secondary purification (reverse osmosis technology or Milli-Q technology), ions with the lowest radii remain in water, including $Na^+$ (0.98 Å) and $Cl^-$ (1.81 Å). These ions are the main impurities of purified water. The authors [22–29] consider the occurrence and evolution of cavitation, avoiding nano-bubbles of dissolved gas in aqueous solutions of salts-babstons, to be the reason for the natural heterogeneity of water, found while studying the cavitation phenomena in water—an abnormal decrease in the threshold of mechanical strength. It is possible that the bubble structures, unavailable to our observations, accumulating ions on its surface, further promote the

formation of microcrystals of NaCl visible in an optical microscope. However, this issue requires a special study.

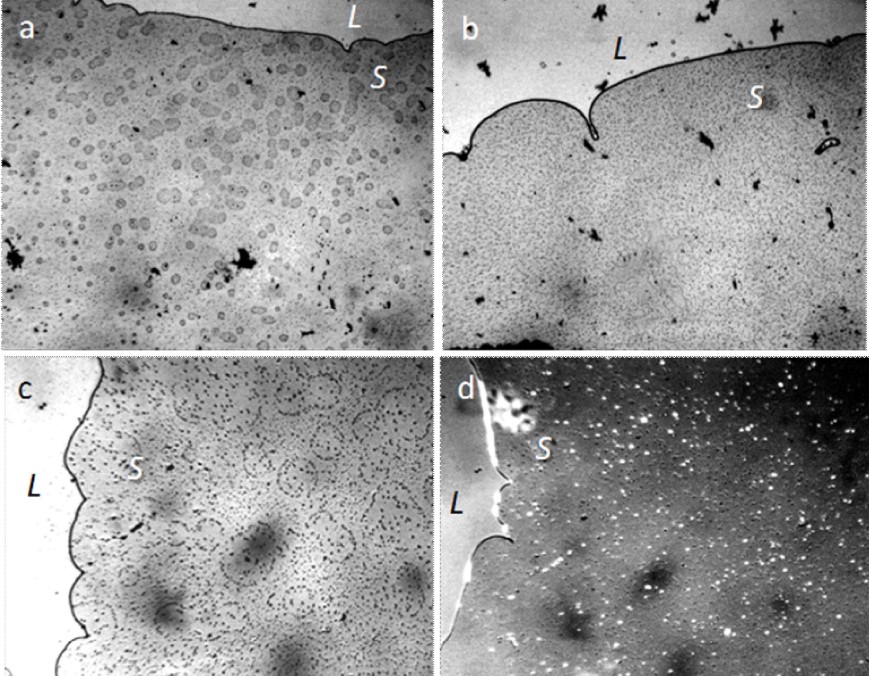

**Figure 17.** Structure of deposits in the drying film of aqueous solutions of NaCl: (**a**) 3%; (**b**) saturated; (**c,d**) 4%. The width of frames (**a,b**) is 3 mm; (**c,d**) 1 mm; (**d**) a picture in a semi-dark field; L is the evaporating liquid phase. The liquid layer thickness is 8 μm.

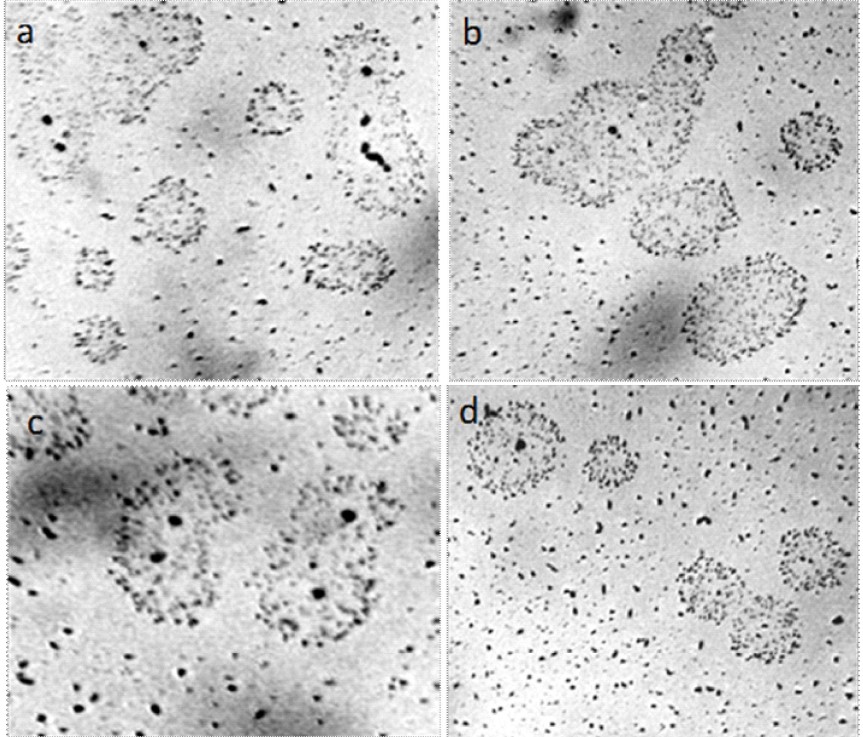

**Figure 18.** Coacervate structures of deposits in a dried film of 3% aqueous NaCl solution. The width of each frame is 0.25 mm, (**a,d**) are different fields of view in the same preparation. The liquid layer thickness is 8 μm.

## 4. Discussion

It is natural to ask why there is so much crystalline salt—sodium chloride—in water, and why it does not dissolve to the ionic state, as strong (nonassociated) electrolytes should [53]. Earlier [45], we suggested that microcrystals of salt found in water can have an aerosol origin. According to the report of the Intergovernmental Panel on Climate Change (IPCC) for 2001, the annual release of sea salt (NaCl with an admixture of $K^+$, $Mg^{2+}$, $Ca^{2+}$, $SO_4{}^{2-}$) from the ocean surface into the atmosphere is 3300 megatons per year [54]. The size of the salt crystals in the atmosphere is, in general, a few microns or less with a predominance of micrometer particles. According to [55], the size of salt crystals in the atmosphere can reach 100 microns. A significant part of NaCl also enters the atmosphere as part of industrial emissions, volcanic activity, vehicular pollution, and human economic activity. The size of the salt crystals in the atmosphere is mostly from fractions to a few microns with a predominance of micrometer particles. Sparging of distilled water by ambient air leads to a significant increase in its conductivity [45].

The existence of a thin film of water on the surface of NaCl crystals was established and confirmed by different research methods [46–51]. The fact that NaCl microcrystals with an average diameter of 0.4 mm at a relative humidity below 50% contain abnormally large amounts of water was experimentally established in [47]. The authors of this study believe that the structure of salt crystals forms cavity-type pockets that are filled with water. The formation of bound water in interpackage space of the crystal lattices of the sliding type has been known for a long time [56]. Given that the mineralogical composition is the same, the solvation shells, formed on the surface of larger particles, are thicker than the ones formed on the surface of the smaller ones. This is due to the different surface curvature and a varying degree of reducing the tension of the force field in proportion to the distance from the particle [56]. It is shown that there is practically no solvent capacity with respect to salts in bound water [57]. There is a concept of bound water being a two-dimensional fluid: It has the property of an ordinary viscous fluid along the surface of the particles and the properties of a solid body in a direction normal to it [58]. The calculation of the hydrate numbers for $Na^+$ ions by the compressibility method has shown much larger values than the ones determined by other methods [53]. This method allows the estimation of the volume of hydration shells: The water molecules located in the hydration shell experience maximum compression under the action of a strong electric field of the ion and therefore the increase in pressure compresses only the liquid portion of the solvent. With increasing thickness of adsorption layers of bound water, its dielectric constant is also increasing [59].

Generalizing the literature data, we can assume that if the microcrystalline NaCl coated with the hydration shell, formed in the atmosphere, falls into liquid water, then it has every chance to maintain its integrity, because the thick hydrated shell will protect it from dissolution. Nevertheless, a lot of questions remain unanswered. One of them is the presence of exactly the same structure in "ultrapure" water (OST 34-70-953.2-88), investigated immediately after depressurization of the plastic container (Figure 6). It should be noted that tightness failure at the opening of the container entails an immediate rapid absorption of atmospheric air, since the initial pressure inside the container is about 50 atm. That is, microspheres with salt crystals can get into the water from the air. On the other hand, another mechanism of microcristal (mesocrystal) formation in solution seems to be possible too. It is so-called nonclassical crystallization—direct self-assembly of crystalline nanoparticles to a micrometer-sized superstructure [60,61]. Besides, two novel droplet phases are shown to exist even in dilute highly charged colloidal suspensions, a liquid-droplet phase and a crystal-droplet phase [62]. But, consideration of these mechanisms is beyond the scope of this paper.

The International Organization for Standardization (ISO) gives the following standards for the ion content in distilled water (Table 1) [63].

**Table 1.** Maximum contaminant levels in purified water (International Organization for Standardization (ISO)).

| Contaminant | Parameter | Grade 1 | Grade 2 | Grade 3 |
|---|---|---|---|---|
| Ions | Resistivity at 25 °C [MΩ·cm] | 10 | 1 | 0.2 |
| | Conductivity at 25 °C [μS·cm$^{-1}$] | 0.1 | 1.0 | 5.0 |

According to the calculations made in [23] (Table 1), the density of the actual dissolved sodium and chlorine ions in water at T = 25 °C corresponds to the values of the specific resistance of the solution. For pure water, the density is in the range from $1.4 \times 10^{13}$ cm$^{-3}$ to $4.5 \times 10^{15}$ cm$^{-3}$. It is recommended to consider solutions with large density values as specially prepared salt solutions based on thoroughly purified water. Using this information we were able to determine the density of Na$^+$ and Cl$^-$ impurity ions in distilled water in our experiment, it was approximately $2.2 \times 10^{16}$ cm$^{-3}$.

If one millimole of NaCl contains $6 \times 10^{20}$ molecules, the weight of which is 0.058 grams, then in our experiment the weight of NaCl molecules in cm$^3$ will be $2.127 \times 10^{-6}$ grams. Therefore, our concentration corresponds to $(2.127 \times 10^{-6})/(0.058) = 3.67 \times 10^{-5}$ mmol. If all molecules had gone into a crystalline state, their volume would have occupied $0.962 \times 10^{-6}$ cm$^3$ (based on the data that the weight of one cm$^3$ of NaCl crystal is 2.21 g). Actually, density fluctuations that occur in aqueous systems can lead to the formation of ionic drops, which can transform to nanocrystals that afterward form microcrystals that we observe in water.

The mass of salt sediment on the glass after drying a drop of distilled water and a drop of tap water of the same volume (3 μL) can be visually assessed by Video S1 and Video S2 (see the Supplementary Materials).

While doing research in highly specialized areas, we often forget about the integrity of the world around us. The water and air elements are in indissoluble unity and are subject to interdependent dynamics. In the natural world, there is no "ultrapure" water, nor highly purified preparations. Our attempt to look at this world has revealed amazing facts about the structure of water at the micro level, its ability to create new phases and the dynamic balance between free and bound states [20,21]. Upon evaporation of free water containing hydrated salt microcrystals (dehydration condensation [64]), coacervates are formed, and it is a process that does not require additional energy inflow. The phenomenon of dehydration condensation was considered by S. Fox as the initial path to the emergence of living cells [65], and the coacervate theory of A.I. Oparin about the origin of life through the appearance of phase boundaries and the redistribution of the components of the solution became widespread in the middle of the last century [66]. The same mechanisms—with the participation of water and ions—undoubtedly work in initiating a nonspecific cell reaction to stimuli [67]: A decrease in the degree of dispersion of colloids (coagulation, coacervation), an increase in the viscosity of protoplasm, or vice versa—its dilution. Interesting facts about the role of water and ions in the function of a living cell are given in G. Ling's book [68]. In particular, the author believes that it is the adsorbed water, rather than the membrane, that plays the role of the diffusion barrier, and the selective binding of the K+ ions to the cells excludes the need for a sodium pump.

According to optical microscopy under natural conditions, we have shown that water is a dispersed system in which the dispersed phase is represented by microcrystals of salt surrounded by thick shells of hydrated water and structures of the next level of hierarchy—coacervates that combine these primary structures into a separate phase. The idea of the possible role of hydration as the leading integration factor in the organization of biosystems at different levels of their hierarchy was first expressed and argued by N.A. Bul'enkov [69]. Based on the correspondence of experimentally established structures of native forms of periodic biopolymers to parametric water structures (by metrics, topology, and symmetry), the author suggested their probable determining role in the chemical evolution of the first biopolymers. It is concluded that the possibility of water molecules forming ideal self-organizing fractals opens the way for self-assembly of biosystems of subsequent hierarchy levels.

We examined the ability of water to form new phases at the micro level with the participation of other natural factors—salts and the process of evaporation of the dispersion medium (liquid water). As our studies have shown, at this level of hierarchy, the constructive tool is not only hydration, but also dehydration. The "elementary microparticle" of the air and water media—salt crystal surrounded by hydrated water—also deserves attention, as a factor combining air and water elements. Answering the question posed at the beginning of the article, we confirmed that at the micro level water is a dispersed system consisting of free (dispersion medium) and polymer (hydrate) constituents in dynamic equilibrium. We hope that the new data shed light on some abnormal properties of water, for example, the rate of change of a number of physical characteristics (thermal conductivity, refractive index, specific conductivity, surface tension) in the temperature range of ~50 °C to 60 °C with a monotonous increase in temperature from 0 °C up to 100 °C (according to the review ([37], page 672)). This phenomenon can be associated with a change in the structuring at the micro level—the destruction of the formed aggregates. This is the topic for future research.

## 5. Conclusions

Using optical microscopy, we confirmed that water and water solutions are, actually, the microdispersed systems in which the size of the dispersed phase is from tens to hundreds of microns. Previously this was noted by other researchers with the help of other tools. The observation of water in a thin (8 μm) layer, whose thickness is of the order of the diameter of the observed structures, made it possible to observe in each of them the presence of a contrasting microparticle located strictly in the center of the sphere. This is novel information about the structures. We have shown experimentally that these microparticles are microcrystals of sodium chloride, and the surrounding spheres are layers of hydrated water. It protects salt crystals from dissolution. When the ionic strength increases, the microdispersed phase coagulates, forming coacervates. Possible ways for sodium chloride to enter the water are external contaminations, but the possibility of the formation of crystals inside the water also cannot be ruled out. Further research is needed to trace the delicate mechanism of the formation of sodium chloride crystals from ionic to microcrystalline state in liquid water.

**Supplementary Materials:** The following are available online at http://www.mdpi.com/2073-4352/9/1/52/s1, Figure S1: Diffraction pattern of crystals formed after evaporation of distilled water, Table S1: The results of chromatography-mass spectrometric analysis of a sample of dried coffee for water content, Video S1: The process of drying a drop of distilled water sitting on a glass substrate, Video S2: The process of drying a drop of tap water sitting on a glass substrate.

**Author Contributions:** Experiments and originated draft preparation: T.Y.; conceptualization—V.Y. & T.Y., supervision, funding acquisition and project administration—V.Y.; manuscript review and editing: V.Y. & T.Y.

**Funding:** This work was supported by the Ministry of Education and Science of Russia (Project No. 14.Y26.31.0022).

**Acknowledgments:** The authors are deeply grateful to their colleagues: D.B. Radishchev, who carried out research using a scanning interference microscope, and A.G. Sanin for the discussion and technical support of the work.

**Conflicts of Interest:** The authors declare no conflict of interest.

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
