# Peer review of "A Study of the Structural Organization of Water and Aqueous Solutions by Means of Optical Microscopy"

_crystals, doi:10.3390/cryst9010052_

Round 1
Reviewer 1 Report
The article 'A study of the structural organization of water and 3 aqueous solutions by means of optical microscopy' highlights the importance of studying the structural organization of water. They have good supporting arguments to prove that the crystals belong to sodium chloride however, they shouldn't rule out the possibility of any crystals, I mean they could be some other isostructural compounds (similar PXRD patterns as NaCl). However, in the present scenario, the given experimental backup is satisfactory. I found that there is some ambiguity about the presence of sodium chloride from the atmosphere, I am wondering is it possible to study the possibility of NaCl or any other contamination's presence in the water before conducting the experiments? also, which technique have you measured the thickness of this layer 8 μm? is it just a microscope?
Author Response
Cover Letter (Resubmission) Manuscript ID: crystals-413101
Dear the Editors,
We represent a new version of our Manuscript after extensive revisions. We tried to fulfill all the requirements of our respected reviewers. Now we send you:
1) Manuscript with notes on the changes made (red text denotes newly written text, and yellow marker indicates small fixes), and
2) The same Manuscript without any remarks.
Reviewers' comments were constructive and allowed us to improve (in our opinion) the quality of the article. Below are the answers to the reviewers in accordance with their sequence numbers.
Dear the Reviewer №1,
Thank you very much for your kindly review. We have corrected the shortcomings of our work in accordance with your comments. However, other reviewers expressed more radical opinions and, in accordance with them, the manuscript was finalized. We hope that the new version will not cause you rejection. Anyway your opinion would be helpful.
Dear the Reviewer №2,
Thank you very much for your opinion on our manuscript. This will help us better formulate the essence of our work.
So what motivated us to write this article?
In short, it was the desire to find out what is water at the micro level and share our findings with colleagues. Millions of people in the world are engaged in research work, one way or another connected with the aquatic environment. Most of them, according to tradition, evaluate it exclusively as a “molecular network” or a set of “flickering” nanoclusters. We made sure that there is a structural organization of water at a higher level of the hierarchy, closer to our world. And this fact does not depend on the properties of the substrate and is manifested even in highly purified water. We consider this sure is important for natural science. The simplicity of the experiment allows everybody to check it themselves who doubts. In other words, our article is phenomenological.
We have shown that the structural unit of the microdispersed phase is a light sphere of ~ 10 microns in size with a contrasting microparticle in the center. After evaporation of the liquid part of the water, the spheres remain on the substrate, and the microparticles begin to grow. After a week they acquire the appearance of crystals. It was confirmed by crystallography that these were sodium chloride crystals.
In order not to overload the reader with information, we divided the results of our work into 3 preprints:
1. A study of structural organization of water and aqueous solutions by means of optical microscopy;
2. Giant water clusters: where are they from? https://arxiv.org/ftp/arxiv/papers/1810/1810.05452.pdf
3. Two-phase water: structural evolution during freezing – thawing according to optical microscopy. https://arxiv.org/ftp/arxiv/papers/1811/1811.06768.pdf
A more detailed study of the origin of NaCl in water is given in the second preprint, which we plan to prepare for printing after the publication of the essence of the phenomenon.
The prehistory that led us to the study of the microstructure of water, in this article, "remained behind the scenes." Perhaps this is wrong. The study of auto-oscillatory processes in a coffee solution as a model of a colloidal liquid led us to this problem [21]. We first identified the relationship between the oscillations of the mechanical characteristics of the drying fluid, the volume of the dispersed phase and the dynamics of growth and destruction of the described microscopic structures, led a mathematical model of these processes [22]. At that time, we could not assume that hydrophilic microparticles surrounded by a thick hydration shell are salt microcrystals. This was revealed only in the study of water and is reflected in this manuscript. Now we are engaged in the automatic registration of self-oscillatory processes in water with different ion concentrations by means of hours-long recording of the electrical characteristics of these liquids. But this is a completely different work.
Dear the Reviewer №3,
Thank you very much for your helpful review. Thanks to your recommendations, our article has become, in our opinion, more logical and meaningful. This should help us in preparing our next publications too.
------------------------------------------------------------------------------------------------------------------------------
Dear the Reviewer,
Thank you very much for your kind support.
All our answers you can find in the attached file
"Response to Reviewer 1 Comments."
We corrected the heading according to the comment made. If you think we should correct the text, please let us know.
Respectfully,
the authors.

Reviewer 2 Report
See attached file

Author Response
Dear Professor Belobrov,
Many thanks for the attentive reading of our work and benevolence. All comments indicated by you are taken into account and corrected.
Respectfully,
the authors.

This manuscript is a resubmission of an earlier submission. The following is a list of the peer review reports and author responses from that submission.
Round 1
Reviewer 1 Report
This paper describes the study of intrinsic property of water (tap water and ultrapure water) and different aqueous solutions using optical microscopy. Authors have used different aqueous solutions using coffee, wine and NaCl to find the structural organization of water.
I think the title should change as ‘A study of the structural organization ……..
Page 1, line 12: The first sentence is not conveying your point correctly please re-write it.
Page 1, line 12: replace microlevel with micro level
Page 1, line 13: I think it is …room temperature conditions instead of room conditions.
Page 5, line 164: replace any more with ‘anymore’
Page 5, line 164: I just wondering whether they have done small-angle X-ray diffraction or some other XRD in order to confirm the NaCl crystals or not? If you have recorded better to put them in the supporting information or in the main manuscript.
Also, please change the unit ‘cm-1 ‘ correctly throughout the document.
I think, this manuscript can be interesting to the ‘Crystals’ readers.
Reviewer 2 Report
This article presents a series of microscope photographs of microparticles contained in water and various aqueous solutions. Efforts were made to identify the source of these particles, including eliminating contamination of their microscope slide surfaces as a possible cause.
I wish to state that I am a crystallographer, and this paper was very much not in my area of expertise. I found it difficult to identify from the introduction what the premise of the paper was, what scientific question was being asked and answered. It was clear that a lot of work had gone in to putting this paper together.
The paper was extremely thoroughly referenced, and the previous literature has very clearly been read and taken on board with great attention to detail, which was a joy to see. I found the photographs clear, and the consistent use of a 3 mm viewing area very helpful to judge relative size increase of the particles.
However, there was no quantification of changes in number or size of particles, which I personally would have found incredibly useful. Individual questions regarding the composition and prevalence of these microparticles seemed to all be answered by referring to prior literature.
Specific recommendation:
Figure 3: Labels should be placed on the x and y axes, and the font of the polynomial equation made bigger so that it is legible.
Reviewer 3 Report
I'm collecting all related notes, questions, suggestions etc. into the one file attached.
